# Acquisitive plants exhibit stronger phenological shifts in response to warming: insights from meta-analysis and long-term monitoring

Kexin Xiong [1], Peter B. Reich [2,3,4], Philippe Ciais [5], Chunyan Lu[1], Huimin Zhou [1,6], Xinxin Wang[1], Josep Peñuelas [7,8], Chaoyang Wu [9,10] ✉ & Huiying Liu [1] ✉

As climate warming accelerates, shifts in plant phenology are reshaping the functioning and stability of terrestrial ecosystems. While the roles of climatic drivers in shaping phenological responses to warming are well established, the influence of intrinsic plant functional traits remains poorly understood. Here, we combine two complementary approaches through a meta-analysis of 124 field warming experiments and an analysis of long-term phenological monitoring networks (CPON and USA-NPN) to evaluate phenological responses to warming across a spectrum of resource-use strategies in seasonally cold biomes. Our meta-analysis demonstrates that resource-acquisitive plants, characterized by higher nutrient concentrations and thinner leaves, show significantly stronger phenological responses to experimental warming. This pattern is observed consistently across both leaf-out in spring and senescence in autumn. These results from meta-analysis are further supported by two long-term observational datasets, which also show more pronounced phenological shifts in acquisitive species under long-term warming. Our findings present a trait-climate integration framework that extends beyond conventional environmental drivers, providing a mechanistic foundation to enhance the accuracy of forecasts for plant responses to climate change.

Terrestrial ecosystems are rapidly changing in response to climatic warming[1–3]. Plant phenology is one of the most sensitive indicators of climatic warming[4,5] and plays important roles in regulating ecosystem carbon, nutrient and water cycling[6,7], biodiversity maintenance[8,9] and ecological stability[10,11]. Extensive research has documented the environmental cues driving plant phenological shifts, including temperature, photoperiod, and precipitation[12–15]. Yet, phenological responses to climate warming, especially the frequently observed species-specific patterns, may arise from a complex interplay between these external drivers and internal plant

traits[16–19]. Plant functional traits reflect fundamental ecological strategies for resource acquisition and growth[20]. Reich et al.[21] and Wright et al.[22] proposed the concept of the "leaf economic spectrum", which characterizes species along a continuum from acquisitive (e.g., high specific leaf area and photosynthetic rates, but short lifespans) to conservative leaf strategies (e.g., low specific leaf area and photosynthetic rates, but long lifespans). Trait-based approaches have significantly advanced understanding of plant responses to environmental change, by uncovering mechanisms behind carbon sequestration, ecosystem resistance, and temporal

stability[23–25]. However, how plant functional traits mediate phenological responses to climate warming remains poorly understood.

Trait-based processes are likely to underpin plant phenological responses to climate warming[16,26,27], with divergent possibilities underpinned by functional differences. In spring, warming often advances leaf-out by accelerating the fulfillment of thermal accumulation requirements[4,28]. Acquisitive species may show greater advancement, as their high foliar nitrogen concentrations support rapid macromolecule synthesis, including Rubisco, thereby enabling earlier initiation of leaf growth in response to warming[29,30] (see Fig. 1b: Hypothesis I). In contrast, conservative species, characterized by thicker leaves and low specific leaf area, may respond by advancing leaf-out even more markedly than acquisitive species due to their stronger resistance to early spring frost[31]. In autumn, warming typically delays leaf senescence by slowing chlorophyll degradation[32]. Acquisitive species may exhibit greater delays in autumnal phenology under warming, due to their higher photosynthetic rates and foliar nitrogen concentrations and sometimes younger leaves (for species with indeterminate growth), which sustain net carbon gain despite shortening photoperiods[25]. However, when warming induces water stress, the delay may become more pronounced in conservative species, which tend to have thicker leaves and lower water requirements, conferring greater drought tolerance[24] (see Fig. 1c: Hypothesis II). Elucidating whether and how plant resource acquisition strategies, as defined by the leaf economics spectrum, regulate phenological responses

to warming is crucial for improving predictions of future phenological shifts.

We combine two complementary approaches through a meta-analysis of 124 global field warming experiments and an analysis with data from two long-term phenological monitoring networks, China Phenological Observation Network (CPON) and USA National Phenology Network (USA-NPN), to position species along a resource-use continuum from acquisitive to conservative. These datasets enable us to explicitly link plant functional traits to phenological responses to climate warming in seasonally cold ecosystems. We test two competing hypotheses; Hypothesis I is that acquisitive species respond more to warming (Fig. 1b), while Hypothesis II is that conservative species respond more to warming (Fig. 1c). Our findings, as described below, provide compelling evidence that acquisitive species show greater phenological responses to warming than conservative species, underscoring the critical role of the leaf economic spectrum in shaping phenological changes under future climate change.

## Results

### The links between leaf traits and phenological responses: meta-analysis

Our meta-analysis showed that warming-induced advances in spring phenology were modestly greater in species with higher foliar nitrogen and phosphorus concentrations, larger leaf area, and lower carbon-to-nitrogen ratios (Supplementary Fig. 1). Shifts in autumnal foliar coloring were also associated with multiple foliar traits. For instance,

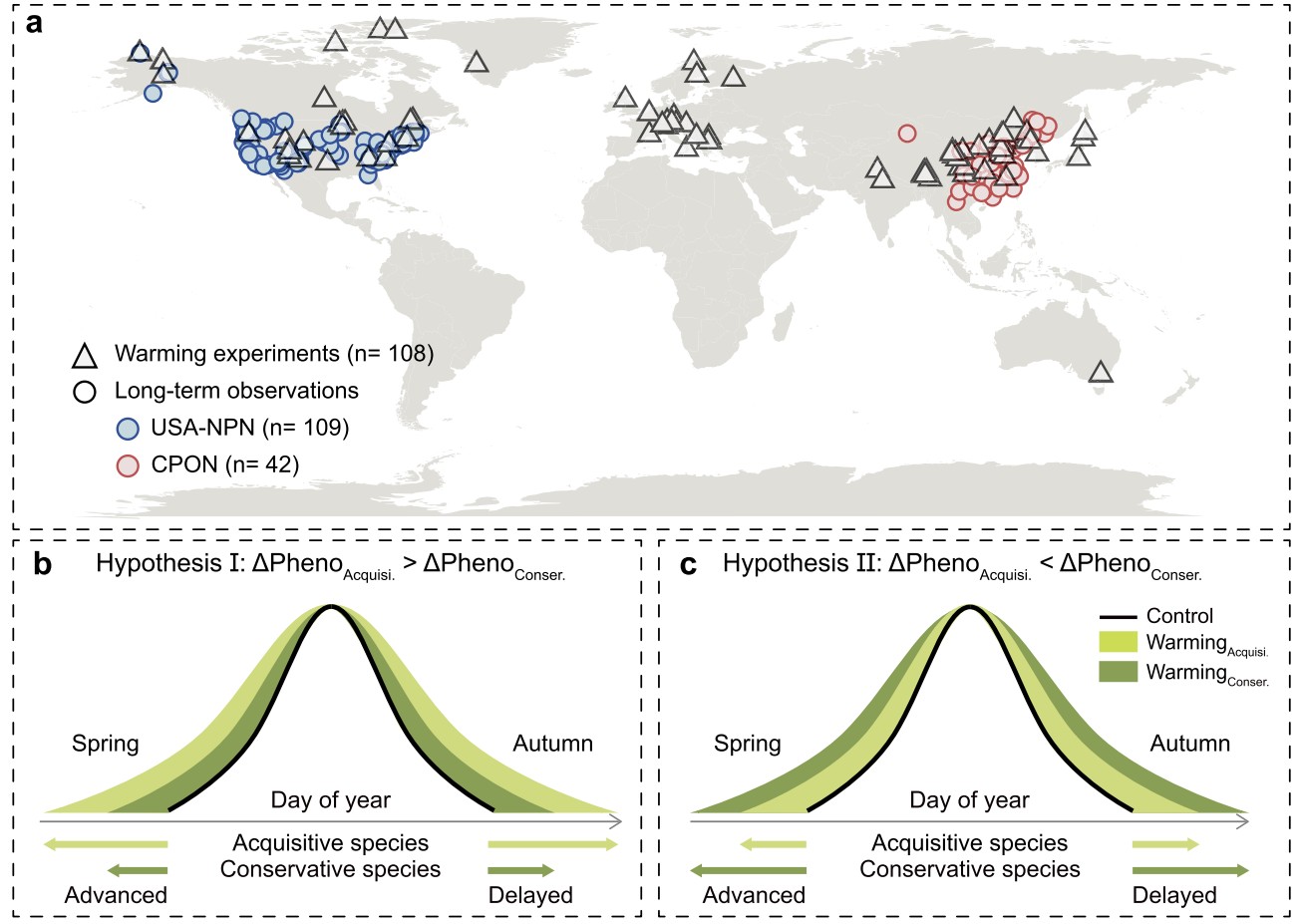

**Fig. 1 | Distribution of the warming experiments and the research hypotheses in this study. a** The distribution of the warming experiments and the long-term ground phenological observations are shown. Gray triangles represent warming experiments, and blue and red squares represent data from the USA National Phenology Network (USA-NPN) and the China Phenological Observation Network (CPON), respectively. **b, c** The hypotheses in this study (more details are in the Introduction). Hypothesis I posits that acquisitive species exhibit stronger phenological shifts. Hypothesis II posits that conservative species exhibit greater phenological shifts. ΔPheno, shifts in foliar phenology; Acquisi., acquisitive species; Conser., conservative species. Source data are provided as a Source Data file.

delays in autumnal foliar coloring were more significant in species with higher foliar nitrogen and phosphorus concentrations, larger leaf area, lower carbon-to-nitrogen ratios, and shorter leaf lifespans (Supplementary Fig. 1). Most of the results remained robust regardless of whether we controlled for confounding climatic factors and experimental variables (see Methods) (Supplementary Fig. 2). These trends were consistent across subgroups based on mean annual temperature, mean annual precipitation, ecosystem type, warming method, warming magnitude, and experimental duration, although the response levels varied (Supplementary Figs. 3-8). Principal component analysis (PCA) reduced the dimensionality of five foliar functional traits. The first principal component (PC1), which explained 54.3% of the total variance, was positively correlated with foliar nitrogen concentrations and specific leaf area, and negatively correlated with foliar dry-matter concentrations and leaf thickness (Fig. 2a and Supplementary Table 1). Species with low PC1 scores (thicker leaves and lower foliar nitrogen concentrations) tended to be resource-conservative, and those with

high PC1 scores (thinner leaves and higher foliar nitrogen concentrations) were resource-acquisitive. Warming-induced shifts in spring phenology were negatively correlated with PC1, and shifts in autumnal phenology were positively correlated with PC1 (Fig. 2b, c). The conclusion remained consistent when we used all ten foliar traits collected in our study (Supplementary Fig. 9). Hierarchical clustering also confirmed that spring and autumnal phenology changed more for acquisitive species but minimally for conservative species (Fig. 2d-f and Supplementary Table 2). The phylogenetic analysis confirmed these patterns, even after controlling for phylogeny (Supplementary Fig. 10 and Supplementary Tables 3,4).

When analyzing the data separately for woody and herbaceous plants, we found that although the impact of functional traits on phenological responses to warming was more pronounced and statistically significant in woody plants, herbaceous plants exhibited a similar directional trend. Specifically, for herbaceous plants, the warming-induced delays in autumn senescence were more evident in

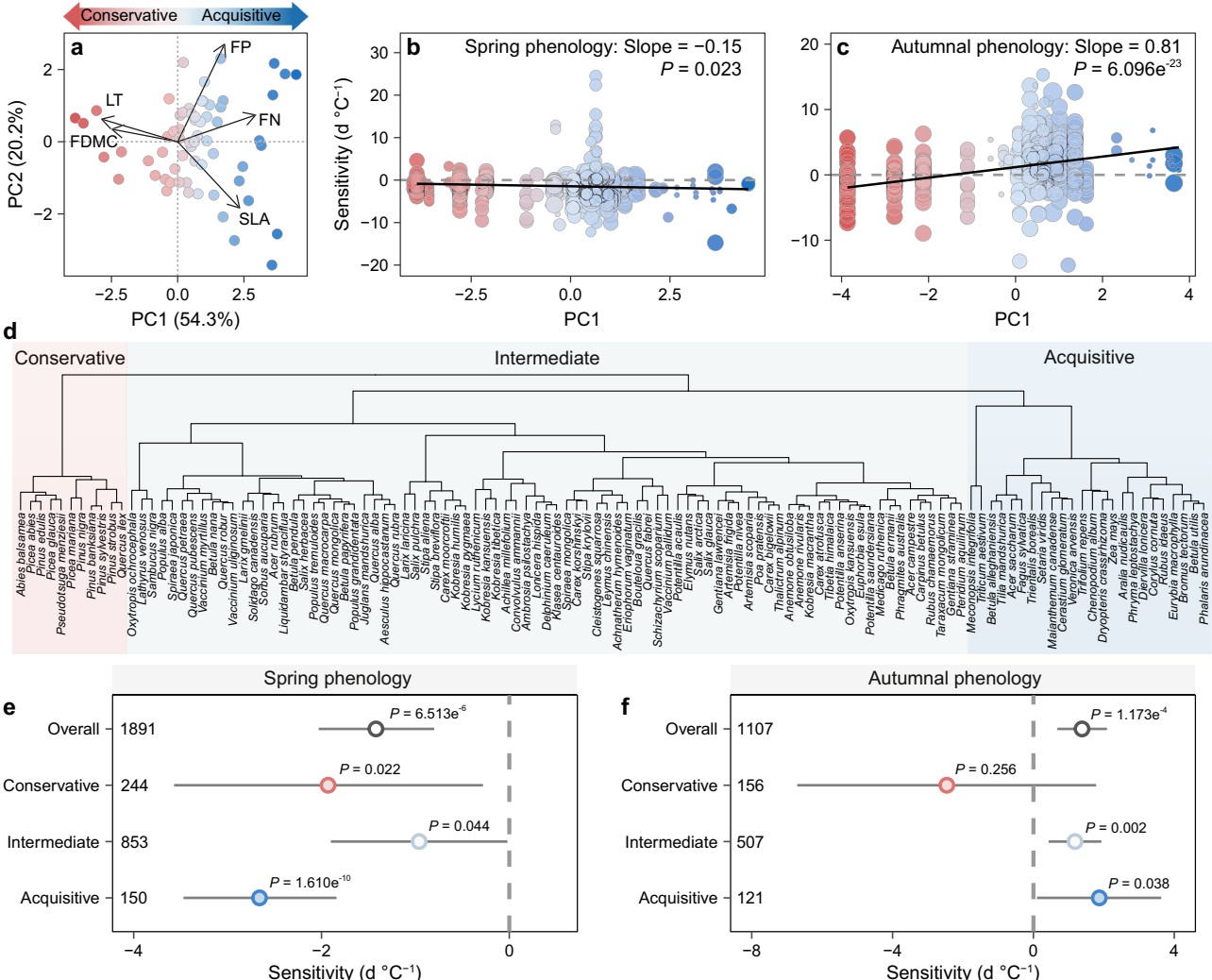

**Fig. 2 | Contrasting phenological responses to warming among species with different resource-use strategies based on experimental manipulations. a** The leaf economic spectrum based on plant traits, where plants with bluer and redder colors are more resource-acquisitive and -conservative, respectively, in their resource use. **b, c** Relationships between the first principal component (PC1) of the economic spectrum and the sensitivities of spring and autumnal phenology to temperature. Solid regression lines indicate significant correlations (*P* < 0.05). The size of the points is proportional to the weight in this meta-analysis. **d**, Classification of species into three functional hierarchical clusters: resource-conservative (pink), intermediate (cyan), and resource-acquisitive (blue). **e, f** Comparison of the

sensitivities of spring and autumnal phenology to temperature across strategy groups. Points with error bars represent mean values and 95% confidence intervals (CIs), with vertical dashed lines representing an effect size of zero. The effects of warming are considered significant if the 95% CIs do not overlap with zero. Statistical significance (*P* < 0.05) was tested using two-sided tests from multi-level meta-analytic linear mixed-effects models without adjustments for multiple comparisons. The numbers on the left denote the sample size. FN, foliar nitrogen concentration; FP, foliar phosphorus concentration; SLA, specific leaf area; FDMC, foliar dry-matter concentration; LT, leaf thickness. Source data are provided as a Source Data file.

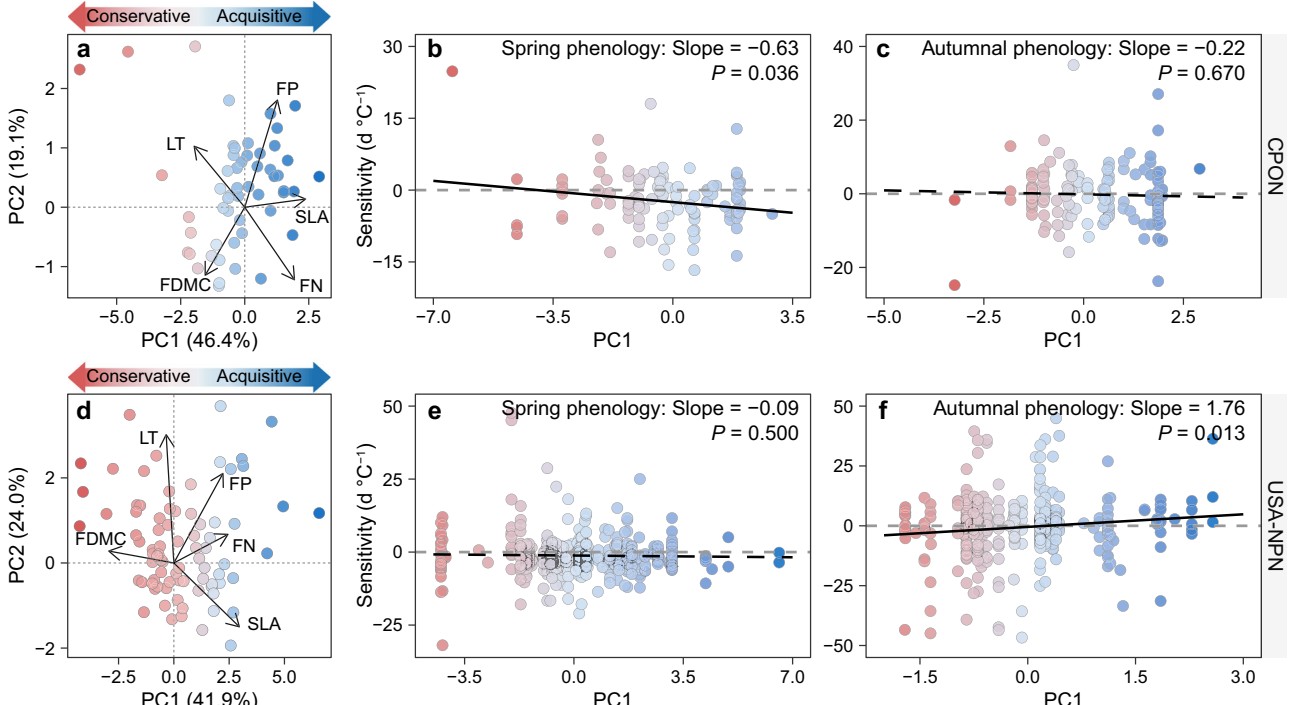

**Fig. 3 | Relationships between foliar economic spectrum and the sensitivity of plant phenology to temperature based on long-term ground observations.** The dataset includes China Phenological Observation Network (CPON) **a**–**c** and the USA National Phenology Network (USA-NPN) **d**–**f**. **a**, **d** Leaf economic spectrum derived from functional traits, with color gradients representing resource-use strategies. The red-to-blue continuum indicates resource-conservative (low SLA, high FDMC) to resource-acquisitive (high SLA, low FDMC) species. **b**, **c**, **e** and **f**, Relationships between leaf economic spectrum principal component (PC1) and the sensitivities of spring and autumnal phenology to temperature. Statistical significance ($P < 0.05$) was tested using two-sided tests from linear mixed-effects models without adjustments for multiple comparisons. Solid regression lines indicate significant correlations ($P < 0.05$). Dashed regression lines denote nonsignificant correlations ($P > 0.05$). FN, foliar nitrogen concentration; FP, foliar phosphorus concentration; SLA, specific leaf area; FDMC, foliar dry-matter concentration; LT, leaf thickness. Source data are provided as a Source Data file.

species with larger leaf areas (Supplementary Fig. 11n). This overall pattern, where acquisitive species (e.g., those with larger leaf areas) showed greater phenological responses, was further supported by PCA for herbaceous plants, even though the results did not reach statistical significance (Supplementary Fig. 11u-w).

### The links between leaf traits and phenological responses: long-term observations

Analyses of long-term phenological data from CPON and USA-NPN indicated that plant traits were key to explain the phenological responses to warming. In the CPON dataset, the advancement of spring phenology was greater in species with thinner leaves and the delay of autumnal phenology was greater in species with larger leaf area (Supplementary Fig. 12). The PCA of the CPON dataset explained 46.4% of the variance, with positive loadings on specific leaf area and negative loadings on leaf thickness (Fig. 3a and Supplementary Table 1). Higher PC1 values indicated stronger acquisitive strategies, and phenological sensitivity in spring was negatively correlated with PC1, indicating that the acquisitive species in China had more pronounced advancement in spring phenology with long-term climate change (Fig. 3b). In the USA-NPN dataset, the advancement of spring phenology was greater in species with thinner leaves and the delay of autumnal phenology was greater in species with higher foliar phosphorus concentrations (Supplementary Fig. 12). The PCA of USA-NPN dataset indicated that PC1 explained 41.9% of the variance, with positive loadings on foliar nitrogen concentrations and specific leaf area and negative loadings on leaf thickness and foliar dry-matter concentrations (Fig. 3d and Supplementary Table 1). Higher PC1 values indicated acquisitive strategies, and lower values indicated conservative strategies. We found that phenological sensitivity in autumn

was positively correlated with PC1, suggesting that acquisitive species also experienced greater delays in autumnal phenology under climate change (Fig. 3f). Plant functional traits generally played a key role in the phenological responses to warming, second only to environmental factors (Supplementary Fig. 13).

### Trait-based forecasting of future plant phenological change

Instead of using a fixed temperature sensitivity obtained from the meta-analysis, we calculated the dynamic phenological sensitivity by parameterizing it with different foliar nitrogen concentrations (Supplementary Fig. 14). Compared to the climate-only projection, our trait-integrated projections forecast smaller phenological shifts in both spring and autumn. Across the two climate scenarios (SSP1-2.6 and SSP5-8.5), incorporating plant traits reduces the projected shifts by 4.00-7.90 days in spring and 1.73-3.45 days in autumn, suggesting systematic overestimation bias in traditional climatically driven projections (Fig. 4, Supplementary Fig. 15 and Supplementary Table 5).

## Discussion

Our meta-analysis, combined with two long-term ground-based datasets, reveals that acquisitive species exhibit stronger phenological responses to climate warming than conservative species, underscoring the pivotal role of plant functional traits, as presented by the leaf economic spectrum, in reshaping phenological patterns. The greater spring advancement and autumn delay in acquisitive species support Hypothesis I. This pattern suggests that acquisitive species may gain a phenological advantage in carbon sequestration under warming, but also face a heightened risk of frost damage[33,34], potentially altering competitive dynamics and influencing species coexistence[35]. Our findings highlight the need to integrate plant functional traits (e.g.,

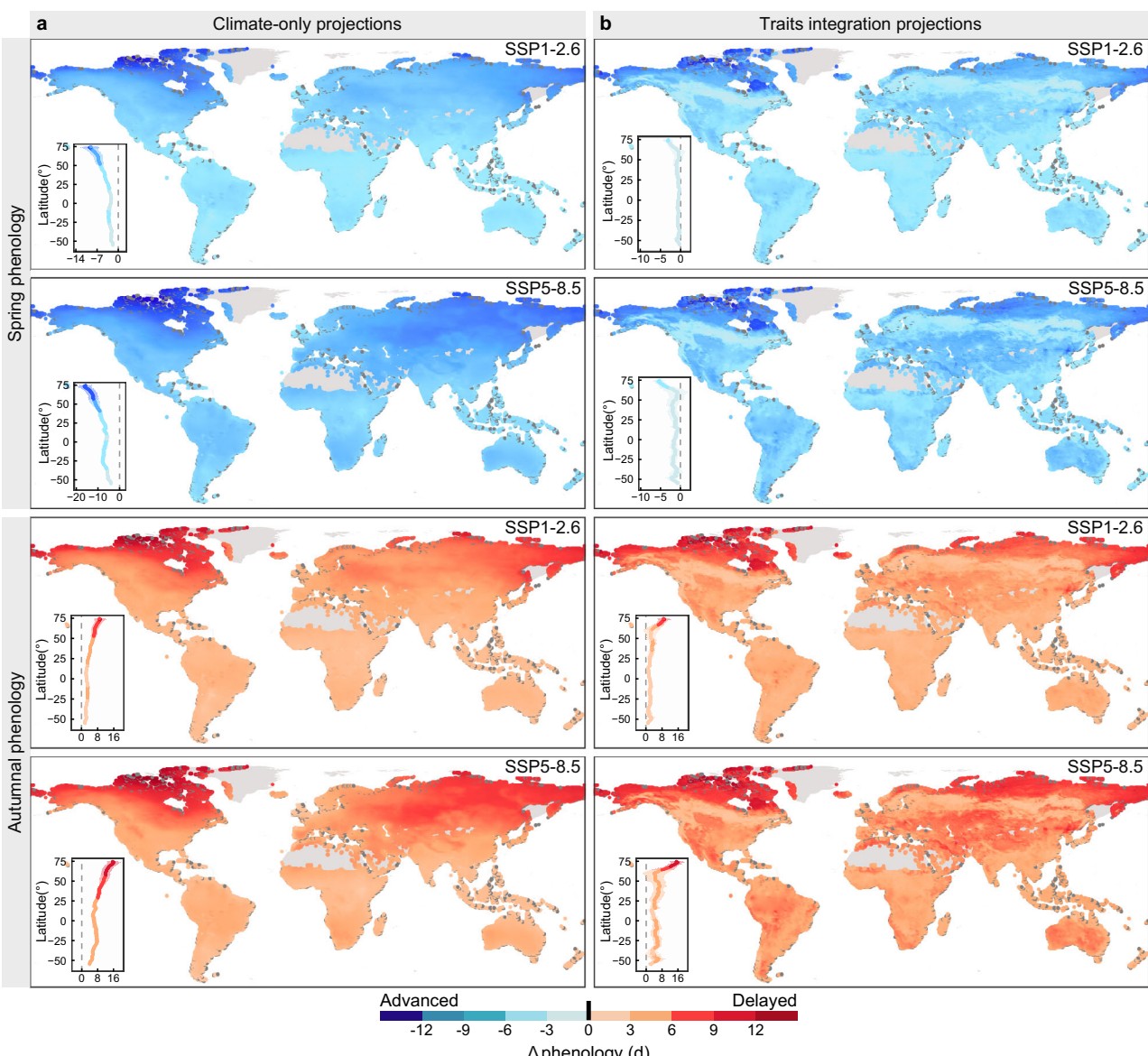

**Fig. 4 | Projections of future foliar phenological changes in terrestrial ecosystems based on climate-only and traits integration projections under two climatic scenarios. a** Projected distribution of potential shifts in spring and autumnal phenology by 2100 under the SSP1-2.6 and SSP5-8.5 scenarios using the climate-only projections. **b** Projected distribution of potential shifts in spring and autumnal phenology by 2100 under the SSP1-2.6 and SSP5-8.5 scenarios using the traits integration projections. Gray indicates areas with missing data. Each panel incorporates the variation of foliar phenological sensitivity across latitudes. SSP, Shared Socioeconomic Paths. Δphenology, shifts in foliar phenology. Source data are provided as a Source Data file.

foliar nitrogen concentrations, specific leaf area), beyond well-established environmental drivers, into predictive models to enhance the accuracy of phenological forecasts under climate change.

The greater phenological sensitivity of acquisitive species to warming that we observed may be attributed to several reasons. First, acquisitive species are characterized by their greater abilities to acquire resources, which enable them to effectively capitalize on the increased ability to acquire resources under warming temperatures, the increased availability of resources due to warmer temperatures[36], or both, thus allowing them to extend their growing season more than conservative species. For instance, climate warming accelerates enzyme-driven reactions and high-nitrogen species, due to their higher nitrogen concentrations, can respond more rapidly to temperature increases[29,30]. In addition, acquisitive species, with higher foliar nitrogen concentrations, are able to accumulate photosynthetic products more rapidly[30,37], providing sufficient resource reserves for

earlier spring phenology under climate warming. Second, acquisitive species are sometimes associated with higher plasticity, enabling them to rapidly adjust to new environmental conditions[25,38]. Acquisitive species under warming may be able to adjust their traits more rapidly, enhancing their ability to cope with changing environmental conditions[39]. In contrast, species with conservative strategies, due to their greater investment in structural development, may incur higher plasticity costs and thus require more time to adapt to new environmental conditions[40]. Third, plants like temperate deciduous woody species need to fulfill both chilling and heat requirements to break endodormancy and ecodormancy, thereby initiating spring leaf-out[4,12]. In many regions, such as subtropical areas, acquisitive species begin accumulating chilling requirements earlier than conservative species, leading to an earlier break in dormancy and an earlier onset of spring phenology[39]. Since acquisitive species start their spring phenology earlier, when temperatures are typically lower and temperature

limitations are more pronounced, this may also explain why they exhibit a stronger response in our study.

We also found that warming delayed the autumnal phenology of acquisitive species but tended to advance that of conservative species. The delays in foliar coloring in acquisitive plants under warming may be attributed to the delayed degradation of chlorophyll and other pigments, maintenance of the activity of the Rubisco enzyme, and the alleviation of constraints of low temperatures at the end of the growing season[32]. This also suggests that in seasonally cold biomes, the autumnal phenology of acquisitive plants may fail to reach carbon sink saturation during autumn under warming condition, contrary to the results from previous studies[41–43]. This could be due to ongoing limitations on photosynthesis imposed by temperature, water, and light conditions in these regions[44,45]. These direct warming effects currently outweigh the indirect effects of warming on an earlier saturation of the carbon sink[46] and the consumption of soil resources[47]. This effect, however, could change as global warming continues, so the saturation of the carbon sink could play an increasingly important role[46]. It is worth noting that if plants possess adaptive capabilities, their resilience to environmental constraints may increase. This could, in turn, reduce the relative importance of carbon sink saturation in the context of future climate change.

We were also surprised to find that experimental warming tended to hasten foliar coloring for conservative species. Conservative plants are typically adapted to resource-scarce environments and may be less plastic[48]. As global warming advanced the leaf-out dates for conservative species, we suspect they were and will be able to adjust their timing of foliar coloring to maintain a relatively stable length of the growing season[13]; however, other conservative evergreen species, such as needle-leafed conifers, will maintain photosynthetic surfaces comprised of multiple leaf cohorts for whatever longer extent the growing season becomes, and do so with larger leaves with shorter leaf-lifetimes[5]. Likewise, conservative species often possess thicker leaves and higher foliar dry-matter concentrations, which come with high construction costs. Previous studies have reported that in autumn, climate warming typically results in carbon losses through respiration, outweighing the carbon gains through photosynthesis[49,50]. As a result, conservative species may opt to end their growing season earlier under warming conditions to conserve more carbon and reabsorb essential nutrients from their leaves, thereby enhancing their ability to adapt to environmental changes[51].

Notably, we found that acquisitive plant species exhibited a stronger phenological response to climate warming, a pattern that was more pronounced in woody plants than in herbaceous plants. Herbaceous plants, particularly perennial grass, tend to store nutrients in underground organs during unfavorable seasons[52]. Consequently, their phenology may be more dependent on the state of these underground storage organs, with weaker relationships to the traits of the above-ground leaves[52]. In contrast, the buds of woody plants are usually exposed to the air and can directly sense temperature changes, whereas herbaceous plant buds are often close to the ground or buried in the soil, where they are buffered by the soil environment. As a result, their responses may be more delayed, and the regulatory effect of traits may be slower. Finally, woody plants generally exhibit stable resource allocation patterns and fixed reproductive cycles (e.g., flowering once a year)[53], while herbaceous plants, especially annual species, have more flexible growth strategies, quickly shifting from vegetative to reproductive growth under favorable conditions, sometimes completing multiple reproductive cycles within a year[54]. This high variability might make it more challenging to capture the role of traits in regulating phenological responses.

Beyond plant growth forms, our study reveals that climate background conditions also modulate how functional traits regulate phenological responses. For instance, functional traits showed minimal effect on autumnal phenology in CPON dataset but a stronger effect in the USA-NPN dataset. In East Asia's monsoon region, synchronized water-heat conditions in spring drive plants to significantly advance spring phenology, possibly maximizing resource use and resulting in a strong spring response to warming[55,56]. However, intrinsic constraints on growing season length might limit autumnal phenological plasticity, potentially weakening the observable regulatory role of traits in the CPON dataset. In contrast, in North America's continental climate, the spring phenological response to warming is comparatively weaker, which could allow plants greater flexibility in adjusting autumnal phenology, and might make trait regulation more detectable[55]. These findings underscore that investigating the regulatory role of traits on phenological responses requires integrating additional factors such as plant growth forms and climatic context.

Most phenological models currently focus primarily on the impact of environmental factors on phenology while overlooking the influence of plant traits[57,58]. Our research found that species-level functional traits regulated the extent of phenological responses to future warming, making it essential to incorporate this factor into phenological simulations. Our trait-explicit projections, integrating future climate scenarios with species trait distribution data, reveal that phenological shifts may be systematically overestimated when the regulatory role of functional traits is omitted. If this prediction is accurate, such overestimations of phenological shifts could lead to inflated projections of ecosystem function changes driven by these shifts, including plant productivity and other related ecological processes[50,59,60]. It is worth emphasizing, however, that our current predictions, which are based solely on temperature effects on phenology, may oversimplify reality and therefore urgently require further refinement using empirical studies with multiple drivers of phenology coupled with process-based models. Our findings thus underscore the need for a more comprehensive approach to phenological modeling, one that integrates both environmental and trait-driven dynamics.

## Methods
### Meta-analysis data compilation
We compiled a data set on the effects of experimental warming on spring (i.e. leaf-out) and autumnal (i.e. leaf coloring) phenology before January 2024. Specifically, relevant peer-reviewed literature was identified through systematic searches of Web of Science, Google Scholar, and the China National Knowledge Infrastructure, using the following key words: (climate change OR warming OR temperature rise OR elevated temperature OR increased temperature) AND (leaf out OR leaf unfold* OR leaf emergence OR bud burst OR burst break OR green-up OR leaf color OR leaf senescence) AND (experiment* OR treatment* OR control*).

We then included studies that met the following criteria: (i) warming experiments were conducted in terrestrial ecosystems, (ii) initial environmental conditions were comparable between control and warming plots, (iii) the method, duration, and magnitude of warming were clearly described, (iv) experimental species were known and indicated, and (v) the timing of phenological events (measured as day of year) under both warming and control treatments, or the phenological shifts induced by warming (in days), along with their sample sizes, were reported. Based on the included studies, phenological data were extracted either directly from tables and appendices or extracted from figures using WebPlotDigitizer version 4.7 (https://apps.automeris.io/wpd/). In total, our screening process (Supplementary Fig. 16) produced 3079 observations (including herbaceous and woody plants) from 124 peer-reviewed articles published, comprising 1941 observations for spring phenology and 1138 observations for autumnal phenology (Fig. 1a).

We gathered information on environmental and experimental conditions from each study, including latitude, longitude, ecosystem type, mean annual temperature (MAT), mean annual precipitation (MAP), warming method, experimental duration, and warming

magnitude. In cases where MAT and MAP were not provided in the articles, we obtained them from the WorldClim database (https://www.worldclim.org/). We provide a comprehensive overview and descriptions of all predictor variables in Supplementary Table 6.

## Long-term ground phenological observations

We constructed the database for the long-term ground monitoring of spring and autumnal phenology using data from the China Phenology Observation Network[61] (CPON, 1982-2018) and the USA National Phenology Network[62] (USA-NPN, 1949-2020). We also considered the dataset from the Pan European Phenological database (PEP725; www.pep725.eu) but did not conduct a detailed analysis due to the small number of species in this dataset, which prevented the formation of a standard economic spectrum. In CPON, we extracted the data (only for woody plants) of leaf-out and leaf senescence with records of no less than 10 years. In USA-NPN, we downloaded all "individual phenometrics" data (including herbaceous and woody plants) with at least 5 years of observations and defined the data as spring or autumnal phenology based on the description of phenophase (Supplementary Table 7). In both datasets, robust statistical methods were used for detecting outliers, with records more than $2.5 \times$ the median absolute deviation excluded[63]. We elaborated on the processing flow of USA-NPN data in detail in Supplementary Fig. 17. The final database encompassed 395 observation sites (30-55°N) and 705 taxa of plants (spanning 114 families and 339 genera), comprising 40098 observations of foliar phenological sequences. Data for mean monthly temperature and mean monthly precipitation for each location were obtained from the CRU TS v.4.08 data set at a spatial resolution of 0.5° (https://crudata.uea.ac.uk/cru/data/hrg/cru_ts_4.08/).

## Plant trait data collection

We quantified trait-mediated phenological responses to warming by systematically compiling a suite of 10 foliar functional traits from the TRY Plant Trait Database (version 5.0, www.try-db.org)[64]. The selected traits, critical for characterizing variation in the leaf economic spectrum were: foliar nitrogen concentration (FN, mg g$^{-1}$), foliar phosphorus concentration (FP, mg g$^{-1}$), foliar nitrogen/phosphorus ratio (FN:FP), leaf area (LA, cm$^2$), specific leaf area (SLA, mm$^2$ mg$^{-1}$), foliar carbon concentration (FC, mg g$^{-1}$), foliar carbon/nitrogen ratio (FC:FN), foliar dry-matter concentration (FDMC, %), leaf thickness (LT, mm), and leaf lifespan (LLS, month). We calculated the mean for each species where the database contained multiple entries for these traits.

## Statistical analysis

**Meta-analysis.** We quantified the effects of warming on spring and autumnal phenology for each observation using phenological sensitivity (d °C$^{-1}$)[65,66], calculated as in Eq. (1):

$$\text{Phenological sensitivity} = \frac{X_w - X_c}{\Delta_T} \tag{1}$$

where $X_w$ and $X_c$ are the phenological timings in the warming and control plots, respectively, and $\Delta_T$ is the temperature difference induced by experimental warming.

Each observation was weighted by its reported sample sizes[67] as follows (Eq. (2)):

$$\text{Wr} = \frac{N_w \times N_c}{N_w + N_c} \tag{2}$$

where Wr is the weight, and $N_w$ and $N_c$ are the sample sizes for the warming and control plots, respectively. Larger values of Wr, derived from larger sample sizes, typically provided more precise effect size estimates and were thus given greater weight in the analysis.

We conducted a hierarchical random-effects meta-analysis to estimate the overall effect sizes of warming on phenology and their 95% confidence intervals. The meta-analytical model was implemented using the 'ram.mv' function from the 'metafor' R package (version 4.4-0), with the variable 'Article ID' included as a random factor to account for potential non-independence among multiple observations reported within the same study[68]. We considered the effects of warming to be significant if the 95% confidence intervals did not overlap with 0. Egger's regression and a fail-safe analysis were used to test the publication bias (Supplementary Table 8).

We also conducted a phylogenetic meta-analysis that incorporated evolutionary information among species to account for the potential influence of phylogenetic relatedness on phenological sensitivity to warming. To do so, we used the following steps. First, we constructed phylogenetic trees for species using the 'V.phylo-Maker' R package[69] (version 0.1.0). Second, the phylogenetic trees were transformed into ultrametric trees based on Grafen's methods in the 'ape' R package[70] (version 5.8-1). Third, we then transformed the ultrametric trees into variance-covariance matrices that represented the phylogenetic relatedness among plant species. Finally, we calculated the sensitivity of species to warming using the phylogenetic meta-analysis by including the corresponding matrix as an additional random factor. We also assessed whether the phenological sensitivity to warming was influenced by phylogenetic relatedness using the "phylosignal" function in the "picante" R package (version 1.8.2) to calculate Blomberg's K metric[71].

We classified species into the three functional types (acquisitive, intermediate, and conservative) based on 10 foliar traits mentioned above using hierarchical clustering in the 'stats' R package (version 4.3.2). Subsequently, we used the hierarchical random-effects meta-analysis and phylogenetic meta-analysis to calculate spring and autumnal phenological sensitivities to temperature for the three functional types species respectively. We also categorized plants into deciduous, evergreen, angiosperms, and gymnosperms and calculated their phenological sensitivities to temperature using the hierarchical random-effects meta-analysis (Supplementary Fig. 10 and Supplementary Table 2).

To analyze the regulatory role of plant functional traits in phenological sensitivity to warming, we first used the 'rma.mv' function to investigate the relationships between the foliar traits and the sensitivity of phenology to temperature, incorporating the 10 foliar traits as moderating variables. Then we performed a principal component analysis (PCA) on 5 foliar traits (FN, FP, SLA, LT and FDMC), standardized using Z-scores, using the 'FactoMineR' package (version 2.9)[72] to capture the underlying plant economic spectrum. We used the 'rcorr' function from the 'Hmisc' package (version 5.2-2) to calculate the correlations between the first principal component from the PCA and 5 foliar traits. Consistent with the methodology applied in the first step, we investigated the relationships between the first principal component from the PCA (PC1) and the sensitivity of phenology to temperature. We also used ecosystem type, MAT, MAP, latitude, warming method, experimental duration, and magnitude of warming as random factors in the hierarchical model to test the robustness of the results. Besides, we categorized environmental and experimental manipulation factors into subsets to identify differences in the relationships between the plant functional traits and the sensitivity of phenology to temperature across various subsets, and we also examined the interaction effects of foliar functional traits and experimental and environmental factors on foliar phenology temperature sensitivity of warming (Supplementary Table 9).

## Long-term ground phenological data analysis

To calculate the sensitivity of spring and autumnal phenology to long-term temperature change, we first used a partial correlation analysis to calculate the optimal preseason length for each species at each

location. The optimal preseason length is the period (with 30-d steps) before the mean leaf-out or foliar coloring date for which the partial correlation coefficient between leaf-out or foliar coloring and air temperature was highest. We then calculated the phenological sensitivity to temperature as the slope of the least-squares regression between the date of leaf-out or foliar coloring and mean air temperature over the preseason for each species at each location. For the long-term data of ground phenological observations, we first used the 'lme' function in the 'nlme' package (version 3.1-163) to analyze the relationship between the sensitivity of spring and autumnal phenology to temperature and 10 foliar traits, with 'Site ID' as a random factor. Then, we used a PCA to analyze the relationship between the 5 foliar traits (FN, FP, SLA, LT and FDMC) for all species and examined the relationship between the sensitivity of spring and autumnal phenology to temperature and PC1. We also calculated the correlations between PC1 and 5 foliar traits. Since USA-NPN is a citizen-science dataset, we screened and reanalyzed the data for at least three observed individuals of the same species in a single site to reduce potential bias in the results (Supplementary Fig. 17).

### Importance of predictors influencing the sensitivity of phenology to temperature

The importance of factors affecting the sensitivity of foliar phenology to temperature was ranked using the 'randomForest' function from the 'randomForest' package (version 4.7-1.2). A regression model with 1000 decision trees was constructed based on the random-forest algorithm. The relative importance of environmental and experimental variables was quantified using two metrics: the permutation importance score and the Gini decrease index. We then used the 'rcorr' function from the 'Hmisc' package to calculate the correlations between the sensitivity of phenology to temperature and the experimental and environmental factors (Supplementary Table 10).

### Projections of future phenological changes

To assess how plant traits would affect the prediction of future changes in foliar phenology, we calculated the shifts of foliar phenology with and without the consideration of plant traits under different climatic scenarios (Supplementary Fig. 14). Initially, when considering plant traits, we refined our model by incorporating foliar nitrogen concentrations. First, we sourced the global distribution of foliar nitrogen concentrations from publicly available data provided by Butler et al.[73]. With this data, we obtained plant phenological sensitivities to foliar nitrogen concentration from this study (Supplementary Fig. 2a,k), which allowed us to calculated how nitrogen levels could influence phenological responses. Next, we collected historical temperature data (1970-2000) and future projections (2081–2100) under two climate scenarios (SSP1-2.6, and SSP5-8.5) from the WorldClim database. Using these temperature data, we predicted future shifts in foliar phenology by integrating the projected temperature changes with both the sensitivity of phenology to temperature and the effects of foliar nitrogen concentrations. However, when plant traits were not considered, we estimated shifts in foliar phenology by multiplying temperature changes by a fixed sensitivity value derived from our meta-analysis. All statistical analyses were conducted in R 4.3.2[74,75].

### Reporting summary

Further information on research design is available in the Nature Portfolio Reporting Summary linked to this article.

## Data availability

The data generated in this study have been deposited in Figshare at https://doi.org/10.6084/m9.figshare.29917052. Plant traits data were obtained from TRY Plant Trait Database (https://www.try-db.org/), climate data from WorldClim (https://www.worldclim.org/) and Climate Research Unit (https://crudata.uea.ac.uk/cru/data/hrg/cru_ts_4. 08/). The long-term ground phenological data of USA National Phenology Network (USA-NPN) are available from the website: https://www.usanpn.org/ results/data. The phenological data from China Phenological Observation Network (CPON) were provided by the Meteorological Information Center of the China Meteorological Administration. Source data are provided with this paper.

## Code availability

Codes for analysis are available in the Figshare repository (https://doi.org/10.6084/m9.figshare.29917052).

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

## Acknowledgements

This research was supported by the National Natural Science Foundation of China (Grant Nos. 32422055 (H.L.), 32130065 (H.L.) and 42125101 (C.W.)) and the National R&D Program of China (Grant No. 2023YFF0806800 (H.L.)). H.L. also acknowledges support from the Shanghai Rising-Star Program (Grant No. 23QA1402900) and Fundamental and Interdisciplinary Disciplines Breakthrough Plan of the Ministry of Education of China (JYB2025XDXM904). We would like to express our gratitude to all the authors of the published papers included in our meta-analysis, as well as to the contributors of the two long-term phenological datasets. We also thank the contributors of the TRY database, as well as the authors who provided the spatial distribution data of the traits used in Fig. 4.

## Author contributions

H.L. and C.W. conceived the study. K.X., H.Z., and H.L. conducted the analysis with inputs from C.L. and X.W. P.R., P.C., J.P., and C.W. provided significant revisions. K.X., C.W., and H.L. wrote the manuscript, with contributions from all co-authors.

## Competing interests

The authors declare no competing interests.

## Additional information

¹Zhejiang Tiantong Forest Ecosystem National Observation and Research Station, Institute of Eco-Chongming, Zhejiang Zhoushan Island Ecosystem Observation and Research Station, School of Ecological and Environmental Sciences, East China Normal University, Shanghai, China. ²Department of Forest Resources, University of Minnesota, St. Paul, MN, USA. ³Hawkesbury Institute for the Environment, Western Sydney University, Penrith, NSW, Australia. ⁴Institute for Global Change Biology and School for Environment and Sustainability, University of Michigan, Ann Arbor, MI, USA. ⁵Laboratoire des Sciences du Climat et de l'Environnement, LSCE/IPSL, CEA-CNRS- UVSQ, Université Paris-Saclay, Gif-sur-Yvette, France. ⁶State Key Laboratory of Grassland Agro-ecosystems, College of Ecology, Lanzhou University, Lanzhou, China. ⁷CREAF, Cerdanyola del Vallès, Barcelona, Catalonia, Spain. ⁸CSIC, Global Ecology Unit, CREAF-CSIC-UAB, Cerdanyola del Vallès, Barcelona, Catalonia, Spain. ⁹The Key Laboratory of Land Surface Pattern and Simulation, Institute of Geographic Sciences and Natural Resources Research, Chinese Academy of Sciences, Beijing, China. ¹⁰University of the Chinese Academy of Sciences, Beijing, China. ✉e-mail: wucy@igsnrr.ac.cn; hyliu@des.ecnu.edu.cn

