## [Transparent Peer Review file · Nature Communications]

Acquisitive plants exhibit stronger phenological shifts in response to warming: insights from meta-analysis and long-term monitoring

Corresponding Author: Dr Huiying Liu

Version 0:

Reviewer comments:

Reviewer #1

(Remarks to the Author)

Overall, this is a strong, well-written paper that brings novel insight into how plant traits play a substantial role in shaping temperature-driven shifts in plant phenology. These insights have strong implications for forecasting, especially in light of dynamic global vegetation models; the authors' findings suggest that to-date, forecasted changes in phenology have been much greater for some plant groups than what may actually play out.

The authors take advantage of both published studies and in situ phenology observations from two continents to arrive at their conclusions. I appreciate this comprehensive and creative approach. My only hesitation is that it feels the authors have not provided enough detail regarding which USA-NPN data they downloaded and in which format, and whether they undertook any post-processing of these data before analyzing. I feel this may impact the reproducibility of the study.

Specific comments:

- I wonder if you might consider a conceptual diagram to support the hypotheses presented in the 2nd paragraph of the Introduction. I found myself wanting to bring reference to Figure 1b,c up to this section of the paper, but realize it encapsulates more scenarios than are described in this part of the Introduction. Keeping track of all of the potential changes, and the underlying mechanisms for them, is a lot; a figure might help here. OR, perhaps you could move reference to Figure 1 up here (reordering panels b & to a & b so that panels are referenced in sequence). Figure 1 really is great - I just crave having it aligned with the mechanisms somehow.
- L99: missing the word "will" before "advance"
- L130: for this reference to PC1 – shouldn't the correlations between PCs and traits be reported in a table? As well, it seems like you could reference Extended Data Fig 1a
- L158: drop "s" from "Figs" here
- L240: "study has" should be "studies have"; consider adding a comma after "respiration"
- L247: add "a" before "study"; perhaps changed "shows" to "showed"
- L249: I don't think you need to say here (again) that conservative species are slower-growing, this has been established multiple times already in the paper
- L257: a readability suggestion: consider changing "the aboveground ones" to "aboveground"
- Please cite the USA-NPN dataset: <https://www.usanpn.org/about/terms#DataUse>, <http://doi.org/10.5066/F78S4N1V>
- Looking at the manuscript text, code, data files on Figshare, and even the reference provided in the Methods for how the USA-NPN dataset was accessed and processed (Wang et al. 2025), I can't tell whether the authors used "status and intensity" or "individual phenometrics" data. I also can't tell if they filtered or processed the data in any other way other than to remove outliers. Given that these data are largely volunteer-contributed, there is the potential that some records could be incorrect, there may be reason to consider the data more carefully. If the authors did implement any sort of filters, it would be very helpful to know.
- L607, Figure 1 legend: "hypothesis" should be "hypotheses"
- I am intrigued by the difference in findings in autumn phenology between CPON and NPN and desired a bit of treatment in the Discussion. Do the authors think these differences may be the result of differences in the duration of the temporal record between the two datasets, in environmental conditions between the continents, in how the data were collected in the two networks, or any other reason?
- Overall, the Introduction and Discussion are very clear and tell a compelling narrative

Supp Info:

- L133, Fig 11: which observational datasets (CPON or NPN) are included in panels c and d? Are these findings from both datasets combined?

Reviewer #2

(Remarks to the Author)

In this manuscript, the authors combine the results of a meta-analysis of warming experiments with long-term ground-based observations to investigate the influence of plant strategies along the acquisitive and conservative continuum on the phenological sensitivity of leaf unfolding and senescence. They found that species with thinner leaves and higher nitrogen concentrations (i.e. acquisitive species) respond stronger to warming including earlier leaf unfolding and later senescence. This has been demonstrated both in experiments and in long-term observations. The authors conclude, and also demonstrate with modelling, that species-specific phenological patterns influenced by traits should be considered in modelling approaches.

Major comment:

1. In many parts in the manuscript, it remains unclear which life forms were considered (trees vs. herbaceous species) to perform the analyses. The phylogenetic analyses indicate that herbaceous species are included, but if I understand correctly, only trees were considered in the analyses with the data from CPON and USA-NPN. The introduction and discussion largely refer to literature on trees, and the pictograms of trees in some graphs also suggest that only trees were analysed in this study. If you want to consciously analyse both life forms, then in my opinion greater emphasis should be placed on highlighting the differences. For me, for example, environmental drivers do not necessarily have the same effect on the phenology of trees and herbaceous species (e.g. winter chilling to break bud dormancy). Furthermore, there is already literature that examines the relationship between phenological sensitivity and traits in herbaceous species (e.g. Bucher and Römermann, 2021; Rauschkolb et al. 2024). Herbaceous species from CPON and USA-NPN should also be considered, or it should be clearly stated that these species are missing from the analyses. In this context, I wonder whether the found patterns are strongly influenced by the fact that herbaceous species, which tend to be acquisitive, and trees, which tend to be conservative, were analysed together. This question arises for me in particular when analysing PC scores. This does not, of course, call the results into question, but I think it is very important that greater attention should be paid to different life forms.

Bucher and Römermann 2021: The timing of leaf senescence relates to flowering phenology and functional traits in 17 herbaceous species along elevational gradients

Rauschkolb et al. 2024: Spatial variability in herbaceous plant phenology is mostly explained by variability in temperature but also by photoperiod and functional traits

2. Methodologically, it remains somewhat unclear to me which studies laid the foundations for your study. I think it would be immensely important to include a list of the original studies in the appendix so that readers can research individual aspects themselves. For example, I wonder how warming experiments with trees were carried out. Were they conducted on plots in forests or on pruned branches or young trees in climate chambers and greenhouses? If these were not field experiments, how does it make sense to take MAT and MAP into account? These questions about methods are certainly also influenced by the lack of clarity about life forms.

Minor comments:

L116: For me, the results of the individual traits and the analyses with the PC axes are somewhat duplicated. This is particularly relevant if one assumes that many of the traits are autocorrelated. You should consider whether you want to show only the analyses with the PC axes in the main document and move the other graphs to the appendix. This would also give you more space to include graphs from the appendix in the main text.

L122: I would write that most of the results remained robust.

L166/Figure 4: I really like the analyses. However, I would like to see the differences of modelling already in the graph. You could only consider two climatic projections but show the graphs with and without the inclusion of foliar nitrogen concentration.

L259: It would certainly be worth mentioning here that you only considered the influence of temperature on phenology for the projections and no other environmental parameters.

L422: Figs. 15 is missing.

L500: I guess Fig. 12 is not the right figure here.

Figure 1: The colours chosen are difficult for red-green colour-blind people like me to distinguish.

Reviewer #3

(Remarks to the Author)

This manuscript combined meta-analysis and two long-term monitoring datasets to show that plant phenological responses to warming depending upon plant functional traits. The results revealed that acquisitive species, with thinner leaves and

higher nitrogen concentrations, showed stronger phenological responses to warming, including earlier leaf-out in spring and delayed senescence in autumn. The findings present a trait-climate integration framework that extends beyond conventional environmental drivers, providing a mechanistic foundation to enhance the accuracy of forecasts for plant responses to climate warming.

It is good to see that two long-term monitoring datasets were used to support their meta-analysis, which confirmed the findings in the study. However, the manuscript needs major revision on the hypotheses and discussion sections.

The major flaw of the study is that the expression of hypotheses are too complicated in the introduction and figure 1. I suggest the author to combine the hypotheses into one: For spring phenology: acquisitive species, characterized by high nitrogen concentrations, advance leaf-out more under warming. For autumnal phenology, acquisitive species show more pronounced advancement of senescence in response to warming, while conservative species exhibit opposite response due to thicker leaves and greater carbohydrate reserves. You do not need to break the hypotheses into hypothesis I and II, scenario A scenario B and so on. You can discuss why the hypothesis is testified or not in the discussion in details. In this case Figure 1 need to be changed, only one hypothesis should be shown. The legends of figure 1 needs to be adjusted accordingly. It is annoying to see too complicated hypotheses and confusing to discern whether figure 1 is the hypotheses or the results.

The discussion can be shorthand and more focused after the hypothesis is adjusted.

Version 1:

Reviewer comments:

Reviewer #1

(Remarks to the Author)

I feel the authors have sufficiently addressed the concerns I raised in my previous review. I appreciate the pains they took to revise text and figures based on my feedback. I have no further concerns; I feel this paper merits publication.

Reviewer #2

(Remarks to the Author)

Thank you for the very comprehensive and successful revision of the paper. My comments have been corrected to my satisfaction. Below are just a few minor suggestions for improvement.

L140: I would delete "also" here.

L197-199: Is this true for all plants or only for trees? Please be more precise here.

Figure 1: I would not change the order of "Acquisitive species" and "Conservative species" in the panels b and c.

Figure 4: Thanks for revising this figure. However, I still see potential for improvements. You are actually interested in the comparison of models with and without considering traits, right? If so, I would create a panel a (spring phenology) and b (autumnal phenology). Left colum of panels could be climate-only projections and the right the projections with the traits. Using this layout it will be easy to compare the two different projections.

Reviewer #3

(Remarks to the Author)

The authors have well responded to reviewers questions, I think the manuscript is ready for publication.

Response letter

We appreciate the three anonymous reviewers' valuable comments and constructive suggestions on the previous version of the manuscript. We have thoroughly addressed all the suggestions made by the reviewers. Below, we list point by point response in the **regular** text next to the all comments/suggestions (in *italics*).

Reviewers' comments to authors

*The line number in this response letter refers to that in the **clean version** of the revised manuscript.

Reviewer #1 (Remarks to the Author):

[R1_1] Overall, this is a strong, well-written paper that brings novel insight into how plant traits play a substantial role in shaping temperature-driven shifts in plant phenology. These insights have strong implications for forecasting, especially in light of dynamic global vegetation models; the authors' findings suggest that to-date, forecasted changes in phenology have been much greater for some plant groups than what may actually play out.

[Response] We thank you very much for your overall positive evaluation and encouraging comments.

[RI_2] The authors take advantage of both published studies and in situ phenology observations from two continents to arrive at their conclusions. I appreciate this comprehensive and creative approach. My only hesitation is that it feels the authors have not provided enough detail regarding which USA-NPN data they downloaded and in which format, and whether they undertook any post-processing of these data before analyzing. I feel this may impact the reproducibility of the study.

[Response] We thank the reviewer for the positive assessment of our study and sincerely apologize for the insufficient details previously provided on the USA-NPN data processing. To address this crucial point regarding reproducibility and data robustness, we have made the following comprehensive changes:

1) We have added a new Supplementary Figure 12 that visually outlines the complete USA-NPN data processing workflow, from initial download to outlier exclusion.

2) We have revised the Methods section (**Lines 463-470**) to provide explicit details on which data types were used ('individual phenometrics'), the criteria for inclusion (e.g., woody plants for CPON, both herbaceous and woody plants for USA-NPN with at least 5 years of observations), and the robust statistical methods employed for outlier detection ($2.5 \times$ median absolute deviation).

3) Recognizing the volunteer-contributed nature of USA-NPN data, we have performed an additional sensitivity analysis to further ensure the reliability of our conclusions. This involved restricting the USA-NPN dataset to sites with at least three observed individuals per species (as depicted in Supplementary Fig. 12, Part II) to minimize potential observational variability and bias. Crucially, this sensitivity analysis reaffirmed our main findings: acquisitive species continued to show significantly greater advances in spring phenology ($P < 0.001$) and marginally greater delays in autumnal phenology ($P = 0.05$) even under these more stringent filtering criteria. These additions significantly enhance the transparency, reproducibility, and robustness of our findings derived from the USA-NPN dataset.

Supplementary Fig.12 Visualization of USA National Phenology Network data processing. In part I, First_Yes_Year, the year of the first “yes” record of the series; First_Yes_DOY, the day of year, ranging from 1 to 366, of the first “yes” record of the series; Last_Yes_Year, the year of the last “yes” record of the series; Last_Yes_DOY, the day of year, ranging from 1 to 366, of the last “yes” record of the series; MAD, median absolute deviation; DOY, day of year. In part II, we screened and reanalyzed the data of at least three observed individuals of the same species in a single site to enhance the robustness of the results. a, The leaf economic spectrum based on five plant traits, where plants with redder and bluer colors are more resource-conservative and -acquisitive, respectively, in their resource use. b, c, Relationships between the first principal component (PC1) of the economic spectrum and the sensitivities of spring and autumnal phenology to temperature. Solid regression lines indicate significant correlations ($P < 0.05$). Dashed regression lines denote nonsignificant correlations ($P > 0.05$). In (a), FN, foliar nitrogen concentration; FP, foliar phosphorus concentration; SLA, specific leaf area; FDMC, foliar dry-matter concentration; LT, leaf thickness. ***, $P < 0.001$.

Specific comments:

[R1_3] I wonder if you might consider a conceptual diagram to support the hypotheses presented in the 2nd paragraph of the Introduction. I found myself wanting to bring reference to Figure 1b,c up to this section of the paper, but realize it encapsulates more scenarios than are described in this part of the Introduction. Keeping track of all of the potential changes, and the underlying mechanisms for them, is a lot; a figure might help here. OR, perhaps you could move reference to Figure 1 up here (reordering panels b & to a & b so that panels are referenced in sequence). Figure 1 really is great - I just crave having it aligned with the mechanisms somehow.

[Response] We thank the reviewer for these suggestions. To enhance the alignment between the text and the figure, we have revised the second paragraph of the introduction and simplified Figure 1 to ensure they present a consistent set of scenarios. The corresponding changes are highlighted in **Introduction (Lines 69-81)**: “In spring,

warming often advances leaf-out by accelerating the fulfillment of thermal accumulation requirements^{4,28}. Acquisitive species may show greater advancement, as their high foliar nitrogen concentrations support rapid macromolecule synthesis, including Rubisco, thereby enabling earlier initiation of leaf growth in response to warming^{29,30} (see Fig.1b: Hypothesis I). In contrast, conservative species, characterized by thicker leaves and low specific leaf area, may respond by advancing leaf-out even more markedly than acquisitive species due to their stronger resistance to early spring frost³¹. In autumn, warming typically delays leaf senescence by slowing chlorophyll degradation³². Acquisitive species may exhibit greater delays in autumnal phenology under warming, due to their higher photosynthetic rates and foliar nitrogen concentrations, which sustain net carbon gain despite shortening photoperiods²⁵. However, when warming induces water stress, the delay may become more pronounced in conservative species, which tend to have thicker leaves and lower water requirements, conferring greater drought tolerance²⁴ (see Fig.1c: Hypothesis II).”

Fig. 1 Distribution of the warming experiments and the research hypotheses in this study. **a**, The distribution of the warming experiments and the long-term ground phenological observations are shown. Gray triangles represent warming experiments, and blue and red squares represent data from the USA National Phenology Network (USA-NPN) and the China Phenological Observation Network (CPON), respectively. **b, c**, The hypotheses in this study. Hypothesis I posits that acquisitive species exhibit stronger phenological shifts. This is attributed to their high foliar nitrogen concentrations and photosynthetic rate, which support rapid macromolecule synthesis and sustained carbon gain, facilitating faster and larger responses to warming. Hypothesis II posits that conservative species exhibit greater phenological shifts based on the idea that their thicker leaves and lower water demand enable them to better tolerate warming-induced water stress. As a result, these species may have a greater capacity to adjust their phenology in

response to climate warming. Δ Pheno, shifts in foliar phenology; Acquisi., acquisitive species; Conser., conservative species.

[R1_4] L99: *missing the word “will” before “advance”*

[Response] We thank for your suggestion, and have revised it as suggested (**Lines 91-93**), which is now quoted as follows: “Hypothesis I predicts acquisitive species, characterized by high nitrogen concentrations, will advance leaf-out more under warming.”

[R1_5] L130: *for this reference to PC1 – shouldn’t the correlations between PCs and traits be reported in a table? As well, it seems like you could reference Extended Data Fig 1a*

[Response] Thanks for mentioning this issue. We supplemented an additional analysis and presented the results in a table (Supplementary Table 1) to provide the correlations between PCs and traits. We further added the description of this analysis in **Methods** as (**Lines 530-532**): “We used the ‘rcorr’ function from the ‘Hmisc’ package (version 5.2-2) to calculate the correlations between the first principal component from the PCA and 5 foliar traits.”

Data source	FN	FP	SLA	LT	FDMC
Meta-analysis data (Figure 2a)	PC1 0.680***	0.568***	0.620***	-0.634***	-0.610***
long-term ground data (Figure 3a)	PC1 0.649***	0.564***	0.905***	-0.738***	-0.631***
long-term ground data (Figure 3d)	PC1 0.634***	0.561***	0.798***	-0.182***	-0.639***

~~Supplementary Table 1. Correlations between the first principal component (PC1) of the economic spectrum and five foliar traits in Figure 2a, Figure 3a and Figure 3d. FN, foliar nitrogen concentration; FP, foliar phosphorus concentration; SLA, specific leaf area; LT, leaf thickness; FDMC, foliar dry-matter concentration; PC1, the first principal component of the economic spectrum. ***, $P < 0.001$.~~

[R1_6] L158: *drop “s” from “Figs” here*

[Response] We have corrected it as suggested (**Line 159**).

[R1_7] L240: *“study has” should be “studies have”; consider adding a comma after “respiration”*

[Response] We have revised them as suggested (**Lines 230-231**).

[R1_8] L247: *add “a” before “study”; perhaps changed “shows” to “showed”*

[Response] Done as suggested (**Lines 267-268**): “a recent study has shown that under field conditions, acquisitive tree species generally grow more slowly²⁴.”

[R1_9] L249: *I don’t think you need to say here (again) that conservative species are slower-growing, this has been established multiple times already in the paper*

[Response] Thanks for mentioning this issue and we revised this part as (**Lines 268270**): “In contrast, conservative tree species often exhibit superior growth performance due to their enhanced ability to tolerate adverse environmental conditions²⁴.”

[RI_10] L257: a readability suggestion: consider changing “the aboveground ones” to “aboveground”

[Response] Thanks for this suggestion and we have replaced “the aboveground ones” with “aboveground” in the revised manuscript (**Line 277**).

[RI_11] Please cite the USA-NPN dataset:

<https://www.usanpn.org/about/terms#DataUse>, <http://doi.org/10.5066/F78S4N1V>

[Response] Thank you for pointing this out. We have now properly cited the USA-NPN dataset as suggested, both in the main text (**Line 460**) and in the reference list. “We constructed the database for the long-term ground monitoring of spring and autumnal phenology using data from the China Phenology Observation Network (CPON, 1982-2018) and the USA National Phenology Network⁶³ (USA-NPN, 19492020).”

[RI_12] Looking at the manuscript text, code, data files on Figshare, and even the reference provided in the Methods for how the USA-NPN dataset was accessed and processed (Wang et al. 2025), I cant’ tell whether the authors used “status and intensity” or “individual phenometrics” data. I also can’t tell if they filtered or processed the data in any other way other than to remove outliers. Given that these data are largely volunteer-contributed, there is the potential that some records could be incorrect, there may be reason to consider the data more carefully. If the authors did implement any sort of filters, it would be very helpful to know.

[Response] We sincerely apologize for the insufficient elaboration on the USA-NPN data processing workflow in our initial submission. To address this, we have now included a detailed schematic (**Supplementary Fig. 12**) outlining the complete data processing steps. Specifically, we used the “individual phenometrics” dataset and applied a robust outlier exclusion criterion (records exceeding 2.5 times the median absolute deviation). This robust outlier exclusion criterion was specifically chosen, following established methods (Wang et al., 2025 and Gu et al., 2022), to effectively mitigate the inherent noise and potential inaccuracies often associated with citizen-science datasets. We have uploaded our code for processing long-term ground phenological data (please refer to the **Code Availability** section).

We are also grateful to the reviewer for highlighting the potential data quality concerns inherent in citizen-science datasets. We fully acknowledge that volunteer-based observations may introduce biases, and we have therefore conducted an additional sensitivity analysis to enhance the reliability of our conclusions. This analysis restricted

the USA-NPN data to sites with at least three observed individuals per species to reduce observational variability (Supplementary Fig. 12). The results reaffirm our main findings which is that acquisitive species show significantly greater advances in spring phenology ($P < 0.001$) and marginally greater delays in autumn phenology ($P = 0.05$). We further supplemented the description of this analysis in **Methods** as (Lines 557559): “Since USA-NPN is a citizen-science dataset, we screened and reanalyzed the data for at least three observed individuals of the same species in a single site to reduce potential bias in the results (Supplementary Fig. 12).”

Supplementary Fig.12 Visualization of USA National Phenology Network data processing. In part I, First_Yes_Year, the year of the first "yes" record of the series; First_Yes_DOY, the day of year, ranging from 1 to 366, of the first "yes" record of the series; Last_Yes_Year, the year of the last "yes" record of the series; Last_Yes_DOY, the day of year, ranging from 1 to 366, of the last "yes" record of the series; MAD, median absolute deviation; DOY, day of year. In part II, we screened and reanalyzed the data of at least three observed individuals of the same species in a single site to enhance the robustness of the results. a, The leaf economic spectrum based on five plant traits, where plants with redder and bluer colors are more resource-conservative and -acquisitive, respectively, in their resource use. b, c, Relationships between the first principal component (PC1) of the economic spectrum and the sensitivities of spring and autumnal phenology to temperature. Solid regression lines indicate significant correlations ($P < 0.05$). Dashed regression lines denote nonsignificant correlations ($P > 0.05$). In (a), FN, foliar nitrogen concentration; FP, foliar phosphorus concentration; SLA, specific leaf area; FDMC, foliar dry-matter concentration; LT, leaf thickness. ***, $P < 0.001$.

[R1_13] L607, Figure 1 legend: “hypothesis” should be “hypotheses

” [Response] We have corrected it as suggested (Line 648).

[R1_14] I am intrigued by the difference in findings in autumn phenology between CPON and NPN and desired a bit of treatment in the Discussion. Do the authors think

these differences may be the result of differences in the duration of the temporal record between the two datasets, in environmental conditions between the continents, in how the data were collected in the two networks, or any other reason?

[Response] We thank the reviewer for this insightful question regarding the differences in autumn phenology between the CPON and USA-NPN datasets. We agree that this discrepancy is an important aspect to explore. Our study found that functional traits showed no significant regulatory effect on autumnal phenology in the CPON dataset, whereas a more pronounced effect was observed in the USA-NPN dataset. This discrepancy likely stems from distinct phenological response strategies shaped by regional climatic backgrounds. In the East Asian monsoon region, synchronized water-heat conditions in spring drive plants to significantly advance spring phenology to maximize resource use, resulting in a strong spring response to warming. However, intrinsic constraints on growing season length may limit the plasticity of autumn phenological adjustments, thereby weakening the observable regulatory role of traits on autumn phenology in the CPON dataset. In contrast, under the continental climate of North America, the spring phenological response to warming is comparatively weaker, allowing plants greater flexibility in adjusting autumn phenology. This expanded window for autumn adjustment may lead to a more detectable influence of functional traits, as evidenced in the USA-NPN dataset.

We have accordingly added a detailed explanation in the **Discussion** section (**Lines 252-264**): “Beyond plant growth forms, our study reveals that climate background conditions also modulate how functional traits regulate phenological responses. For instance, functional traits showed minimal effect on autumnal phenology in CPON dataset but a stronger effect in the USA-NPN dataset. In East Asia’s monsoon region, synchronized water-heat conditions in spring drive plants to significantly advance spring phenology, possibly maximizing resource use and resulting in a strong spring response to warming^{55,56}. However, intrinsic constraints on growing season length might limit autumnal phenological plasticity, potentially weakening the observable regulatory role of traits in the CPON dataset. In contrast, in North America’s continental climate, the spring phenological response to warming is comparatively weaker, which could allow plants greater flexibility in adjusting autumnal phenology, and might make trait regulation more detectable⁵⁵. These findings underscore that investigating the regulatory role of traits on phenological responses requires integrating additional factors such as plant growth forms and climatic context.”

This new discussion not only provides valuable nuance to our findings but also underscores the novel insights gained from combining geographically distinct datasets, highlighting the critical context-dependency of trait-phenology relationships and enriching our understanding of how environmental conditions modulate the regulatory role of functional traits.

[R1_15] Overall, the Introduction and Discussion are very clear and tell a compelling narrative

[Response] We thank you very much for your overall positive evaluation and many constructive comments.

Supp Info:

[R1_16] L133, Fig 11: which observational datasets (CPON or NPN) are included in panels c and d? Are these findings from both datasets combined?

[Response] Sorry for the confusion. These findings are the result of the combination of two datasets (CPON and NPN). We have revised the corresponding figure legend and added data labels to the figure (Supplementary Fig. 10).

Supplementary Fig.10 Ranking of key drivers influencing foliar phenological temperature sensitivity using %IncMSE based on random forest analysis. a, b, Ranking of factors influencing spring and autumnal

phenological sensitivity in warming experiments. For meta-analysis data, only field experiments data is included in the analysis. **c, d**, Ranking of factors influencing phenological sensitivity based on two long-term ground phenological datasets. FN, foliar nitrogen concentration; FP, foliar phosphorus concentration; FN:FP, foliar nitrogen/phosphorus ratio; LA, leaf area; SLA, specific leaf area; FC: foliar carbon concentration; FC:FN, foliar carbon/nitrogen ratio; FDMC, foliar dry-matter concentration; LT, leaf thickness; LLS, leaf lifespan; MAT, mean annual temperature; MAP, mean annual precipitation; Ecotype, ecosystem type.

Reference

Wang, J. *et al.* Late spring frost delays tree spring phenology by reducing photosynthetic productivity. *Nat. Clim. Chang.* **15**, 201–209 (2025).

Gu, H. *et al.* Warming-induced increase in carbon uptake is linked to earlier spring phenology in temperate and boreal forests. *Nat. Commun.* **13**, 3698 (2022).

Reviewer #2 (Remarks to the Author):

[R2_1] In this manuscript, the authors combine the results of a meta-analysis of warming experiments with long-term ground-based observations to investigate the influence of plant strategies along the acquisitive and conservative continuum on the phenological sensitivity of leaf unfolding and senescence. They found that species with thinner leaves and higher nitrogen concentrations (i.e. acquisitive species) respond stronger to warming including earlier leaf unfolding and later senescence. This has been demonstrated both in experiments and in long-term observations. The authors conclude, and also demonstrate with modelling, that species-specific phenological patterns influenced by traits should be considered in modelling approaches.

[Response] Thank you very much for this concise and accurate summary of our work, and for acknowledging the methods we used. We have carefully gone through your comments, and addressed each of them accordingly. For more details, please see our point-to-point response to your comments below.

Major comment:

[R2_2] In many parts in the manuscript, it remains unclear which life forms were considered (trees vs. herbaceous species) to perform the analyses. The phylogenetic analyses indicate that herbaceous species are included, but if I understand correctly, only trees were considered in the analyses with the data from CPON and USA-NPN. The introduction and discussion largely refer to literature on trees, and the pictograms of trees in some graphs also suggest that only trees were analysed in this study. If you want to consciously analyse both life forms, then in my opinion greater emphasis should be placed on highlighting the differences. For me, for example, environmental drivers do not necessarily have the same effect on the phenology of trees and herbaceous species (e.g. winter chilling to break bud dormancy). Furthermore, there is already literature that examines the relationship between phenological sensitivity and traits in herbaceous species (e.g. Bucher and Römermann, 2021; Rauschkolb et al. 2024). Herbaceous species from CPON and USA-NPN should also be considered, or it should be clearly stated that these species are missing from the analyses. In this context, I wonder whether the found patterns are strongly influenced by the fact that herbaceous species, which tend to be acquisitive, and trees, which tend to be conservative, were analysed together. This question arises for me in particular when analysing PC scores. This does not, of course, call the results into question, but I think it is very important that greater attention should be paid to different life forms.

Bucher and Römermann 2021: The timing of leaf senescence relates to flowering phenology and functional traits in 17 herbaceous species along elevational gradients

Rauschkolb et al. 2024: Spatial variability in herbaceous plant phenology is mostly

explained by variability in temperature but also by photoperiod and functional traits

[Response] We sincerely thank the reviewer for raising these important points regarding the clarification of plant life forms in our analyses. In the revised version, we have explicitly stated the life-form composition of each dataset in the **Methods** section (**Line 448 and Lines 464-466**): the meta-analysis included both herbaceous and woody plants, the CPON dataset contained only woody plants, and the USA-NPN dataset encompassed both herbaceous and woody species, with 295 herbaceous species and 371 woody species.

In addition, we acknowledge that herbaceous and woody plants may exhibit differential responses to environmental drivers. Accordingly, we conducted separate analyses for these two life forms (**Supplementary Fig. 9**). The results demonstrate that functional traits influence phenological responses in both herbaceous and woody plants, but the strength of this relationship differs markedly between them. For instance, although single-trait analysis indicated that herbaceous species with larger leaf areas showed more pronounced delays in autumn senescence under warming, the overall regulatory effect of traits on phenological responses to warming was stronger in woody plants. We propose that this divergence arises from fundamental physiological and ecological distinctions. Herbaceous plants, particularly perennials, often rely on belowground nutrient reserves, which may buffer their phenology from direct influences of aboveground traits and atmospheric conditions. In contrast, woody plants possess aerial buds that respond more directly to temperature changes. Moreover, woody species typically follow fixed annual growth cycles, whereas herbaceous plants, especially annuals, may display more flexible, multi-cyclical growth patterns. These characteristics complicate trait-based predictions of phenological changes in herbaceous species.

We have added a description of the analysis results in the **Results** section as (**Lines 131-138**): “When analyzing the data separately for woody and herbaceous plants, we found that although the impact of functional traits on phenological responses to warming was more pronounced and statistically significant in woody plants, herbaceous plants exhibited a similar directional trend. Specifically, for herbaceous plants, the warming-induced delay in autumn senescence was more evident in species with larger leaf areas (**Supplementary Fig. 9n**). This overall pattern, where acquisitive species (e.g., those with larger leaf areas) showed greater phenological responses, was further supported by PCA for herbaceous plants, even though the results did not reach statistical significance (**Supplementary Fig. 9u-w**).”

Correspondingly, in the **Discussion** section, we have added the possible reasons for the differences between the phenological responses of woody and herbaceous plants as (**Lines 236-250**): “Notably, we found that acquisitive plant species exhibited a stronger

phenological response to climate warming, a pattern that was more pronounced in woody plants than in herbaceous plants. Herbaceous plants, particularly perennial grass, tend to store nutrients in underground organs during unfavorable seasons⁵². Consequently, their phenology may be more dependent on the state of these underground storage organs, with weaker relationships to the traits of the above-ground leaves⁵². In contrast, the buds of woody plants are usually exposed to the air and can directly sense temperature changes, whereas herbaceous plant buds are often close to the ground or buried in the soil, where they are buffered by the soil environment. As a result, their responses may be more delayed, and the regulatory effect of traits may be slower. Finally, woody plants generally exhibit stable resource allocation patterns and fixed reproductive cycles (e.g., flowering once a year)⁵³, while herbaceous plants, especially annual species, have more flexible growth strategies, quickly shifting from vegetative to reproductive growth under favorable conditions, sometimes completing multiple reproductive cycles within a year⁵⁴. This high variability might make it more challenging to capture the role of traits in regulating phenological responses.”

Additionally, we found the literature you recommended to be highly insightful for understanding how functional traits shape the phenological responses of herbaceous plants to climate change. Accordingly, we have incorporated these references into the **Introduction** section (**Line 58 and Line 69**).

Supplementary Fig.9 Relationships between foliar functional traits and foliar phenology temperature sensitivity based on experimental manipulations under different plant life forms. The foliar phenology includes spring phenology (a–j) and autumnal phenology (k–t). Regression lines are shown when $P < 0.05$. The size of the points is proportional to the weight in this meta-analysis. **u, x**, Leaf economic spectrum derived from functional traits, with color gradients representing resource-use strategies. The red-to-blue continuum indicates resource-conservative (low SLA, high FDMC) to resource-acquisitive (high SLA, low FDMC) species. **v, w, y** and **z**, Relationships between leaf economic spectrum principal component (PC1) and the sensitivities of spring and autumnal phenology to temperature. Solid regression lines indicate significant correlations ($P < 0.05$). Dashed regression lines denote nonsignificant correlations ($P > 0.05$). FN, foliar nitrogen concentration; FP, foliar

phosphorus concentration; FN:FP, foliar nitrogen/phosphorus ratio; LA, leaf area; SLA, specific leaf area; FC: foliar carbon concentration; FC:FN, foliar carbon/nitrogen ratio; FDMC, foliar dry-matter concentration; LT, leaf thickness; LLS, leaf lifespan. *, $P < 0.05$; **, $0.001 < P < 0.01$; ***, $P < 0.001$.

[R2_3] Methodologically, it remains somewhat unclear to me which studies laid the foundations for your study. I think it would be immensely important to include a list of the original studies in the appendix so that readers can research individual aspects themselves. For example, I wonder how warming experiments with trees were carried out. Were they conducted on plots in forests or on pruned branches or young trees in climate chambers and greenhouses? If these were not field experiments, how does it make sense to take MAT and MAP into account? These questions about methods are certainly also influenced by the lack of clarity about life forms.

[Response] We thank the reviewer for these suggestions. Based on your suggestions, we have supplemented a list (please see Supplementary Table 6) of all the original studies included in the meta-analysis in the Supplementary file. Meanwhile, we have also supplemented the specific research subjects and warming methods adopted in each study (please refer to the Data availability section). We agree with the reviewer's point that if it is not a field experiment, taking MAT and MAP into account may not make sense. Therefore, we classified the experimental types (field/controlled) for all studies based on the experimental methods. Correspondingly, we modified all analyses considering MAT, MAP and ecosystem types (Supplementary Tables 10 and 11, Supplementary Figs.1, 5, 6, 7 and 10). In these analyses, only field experiments data is included and the overall results remained the same.

Supplementary Fig.1 Relationship between foliar functional traits and foliar phenology temperature sensitivity based on experimental manipulations after excluding environmental and experimental factors. Only field experiments data is included in the analysis. The foliar phenology includes spring phenology (a–j) and autumnal phenology (k–t). u, The leaf economic spectrum based on five plant traits, where plants with redder and bluer colors are more resource-conservative and -acquisitive, respectively, in their resource use. v, w, Relationships between the first principal component (PC1) of the economic spectrum and the sensitivities of spring and autumnal phenology to temperature after excluding environmental and experimental factors. Regression lines are shown when $P < 0.05$. The size of the points is proportional to the weight in this meta-analysis. FN, foliar nitrogen concentration; FP, foliar phosphorus concentration; FN:FP, foliar nitrogen/phosphorus ratio; LA, leaf area; SLA, specific leaf area; FC: foliar carbon concentration; FC:FN, foliar carbon/nitrogen ratio; FDMC, foliar dry-matter concentration; LT, leaf thickness; LLS, leaf lifespan. *, $P < 0.05$; **, $0.001 < P < 0.01$; ***, $P < 0.001$.

Minor comments:

[R2_4] L116: For me, the results of the individual traits and the analyses with the PC axes are somewhat duplicated. This is particularly relevant if one assumes that many of the traits are autocorrelated. You should consider whether you want to show only the analyses with the PC axes in the main document and move the other graphs to the

appendix. This would also give you more space to include graphs from the appendix in the main text.

[Response] We thank the reviewer for these constructive suggestions. We have moved the results of the individual traits in the meta-analysis and the long-term ground phenological data analysis to the **Extended Data Fig** section, and rearranged Figure 2 and Figure 3.

Fig. 2 Contrasting phenological responses to warming among species with different resource-use strategies based on experimental manipulations. **a**, The leaf economic spectrum based on plant traits, where plants with bluer and redder colors are more resource-acquisitive and -conservative, respectively, in their resource use. **b**, **c** Relationships between the first principal component (PC1) of the economic spectrum and the sensitivities of spring and autumnal phenology to temperature. Solid regression lines indicate significant correlations ($P < 0.05$). The size of the points is proportional to the weight in this meta-analysis. **d**, Classification of species into three functional hierarchical clusters: resource-conservative (pink), intermediate (cyan), and resource-acquisitive (blue). **e**, **f**, Comparison of the sensitivities of spring and autumnal phenology to temperature across strategy groups. Error bars indicate 95% confidence intervals (CIs), with vertical dashed lines representing an effect size of zero. The effects of warming are considered significant if the 95% CIs do not overlap with zero. FN, foliar nitrogen concentration; FP, foliar phosphorus concentration; SLA, specific leaf area; FDMC, foliar dry-matter concentration; LT, leaf thickness. *, $P < 0.05$; **, $0.001 < P < 0.01$; ***, $P < 0.001$.

Fig. 3 Relationships between foliar economic spectrum and the sensitivity of plant phenology to temperature based on long-term ground observations. The dataset includes China Phenological Observation Network (CPON) (a-c) and the USA National Phenology Network (USA-NPN) (d-f). **a, d**, Leaf economic spectrum derived from functional traits, with color gradients representing resource-use strategies. The red-to-blue continuum indicates resource-conservative (low SLA, high FDMC) to resource-acquisitive (high SLA, low FDMC) species. **b, c, e** and **f**, Relationships between leaf economic spectrum principal component (PC1) and the sensitivities of spring and autumnal phenology to temperature. Solid regression lines indicate significant correlations ($P < 0.05$). Dashed regression lines denote nonsignificant correlations ($P > 0.05$). FN, foliar nitrogen concentration; FP, foliar phosphorus concentration; SLA, specific leaf area; FDMC, foliar dry-matter concentration; LT, leaf thickness. *, $P < 0.05$.

[R2_5] L122: I would write that most of the results remained robust.

[Response] We thank for your suggestion, and have revised it as suggested (**Lines 111-113**), which is now quoted as follows: “Most of the results remained robust regardless of whether we controlled for confounding climatic factors and experimental variables (see Methods) (**Supplementary Fig. 1**).”

[R2_6] L166/Figure 4: I really like the analyses. However, I would like to see the differences of modelling already in the graph. You could only consider two climatic projections but show the graphs with and without the inclusion of foliar nitrogen concentration.

[Response] Again, thank for your great suggestions. We have revised Figure 4 as you suggested. We simultaneously presented the results of projections of future foliar phenological changes in terrestrial ecosystems based on climate-only and traits integration projections under SSP1-2.6 and SSP5-8.5 (**Fig.4**). We also revised the result description of Figure 4 in the **Results** section as (**Lines 161-168**): “Instead of using a fixed temperature sensitivity obtained from the meta-analysis, we dynamically calculated the phenological sensitivity by parameterizing it with different foliar nitrogen concentrations (**Extended Data Fig. 4**). Compared to the climate-only

projection, our trait-integrated projections significantly attenuated forecasted shifts in both spring and autumn. In two different climate change scenarios (SSP1-2.6 and SSP5-8.5), trait integration attenuated forecasted shifts by 4.00-7.90 d (spring) and 1.73-3.45 d (autumn), identifying systematic overestimation bias in traditional climatically driven projections (Fig. 4, Extended Data Fig. 5 and Supplementary Table 5).”

Fig. 4 Projections of future foliar phenological changes in terrestrial ecosystems based on climate-only and traits integration projections under two climatic scenarios. **a, b,** Projected distribution of potential shifts in spring and autumnal phenology by 2100 under the SSP1-2.6 (**a**) and SSP5-8.5 (**b**) scenarios using the climate-only projections. **c, d,** Projected distribution of potential shifts in spring and autumnal phenology by 2100 under the SSP1-2.6 (**c**) and SSP5-8.5 (**d**) scenarios using the traits integration projections. Gray indicates areas with missing data. Each panel incorporates the variation of foliar phenological sensitivity across latitudes. SSP, Shared Socioeconomic Paths. Δ phenology, shifts in foliar phenology.

Visualization of future phenological change projections

Step 1: Calculate ΔT

Step 2: Calculate ST

Step 3: Calculate Δ phenology

Extended Data Fig. 4 Visualization of future phenological change projections. SSP, Shared Socioeconomic Paths. ΔT , The changes of temperature. ST, phenological temperature sensitivity. FN, foliar nitrogen concentration. Δ phenology, shifts in foliar phenology.

Extended Data Fig. 5 Future spring and autumnal phenological changes based on climate-only and traits integration projections by 2100 under two climatic scenarios. SSP, Shared Socioeconomic Paths. Δ phenology, shifts in foliar phenology.

[R2_7] L259: *It would certainly be worth mentioning here that you only considered the influence of temperature on phenology for the projections and no other environmental parameters.*

[Response] Thanks for mentioning this issue, and have revised it as suggested (Lines 285-287), which is now quoted as follows: “It is worth emphasizing that our current predictions, which are based solely on temperature effects on phenology, may

oversimplify reality and therefore urgently require validation using process-based models.”

[R2_8] L422: Figs. 15 is missing.

[Response] Sorry for the mistake. We have added the correct diagram here (Line 447), and we have now thoroughly revised the original manuscript to ensure that the manuscript matches all the figures.

[R2_9] L500: I guess Fig. 12 is not the right figure here.

[Response] We apologize again for this mistake. We have corrected the corresponding Figure (Line 524).

[R2_10] Figure 1: The colours chosen are difficult for red-green colour-blind people like me to distinguish.

[Response] We thank the reviewer for mentioning this issue, and we have revised the color in Figure 1 and used different symbols to distinguish data from different sources.

Fig. 1 Distribution of the warming experiments and the research hypotheses in this study. **a**, The distribution of the warming experiments and the long-term ground phenological observations are shown. Gray triangles represent warming experiments, and blue and red squares represent data from the USA National Phenology Network (USA-NPN) and the China Phenological Observation Network (CPON), respectively. **b, c**, The hypotheses in this study. Hypothesis I posits that acquisitive species exhibit stronger phenological shifts. This is attributed to their high foliar nitrogen concentrations and photosynthetic rate, which support rapid macromolecule synthesis and sustained carbon gain, facilitating faster and larger responses to warming. Hypothesis II posits that conservative species exhibit greater phenological shifts based on the idea that their thicker leaves and lower water demand enable them to better tolerate

warming-induced water stress. As a result, these species may have a greater capacity to adjust their phenology in response to climate warming. Δ Pheno, shifts in foliar phenology; Acquisi., acquisitive species; Conser., conservative species.

Reviewer #3 (Remarks to the Author):

[R3_1] This manuscript combined meta-analysis and two long-term monitoring datasets to show that plant phenological responses to warming depending upon plant functional traits. The results revealed that acquisitive species, with thinner leaves and higher nitrogen concentrations, showed stronger phenological responses to warming, including earlier leaf-out in spring and delayed senescence in autumn. The findings present a trait-climate integration framework that extends beyond conventional environmental drivers, providing a mechanistic foundation to enhance the accuracy of forecasts for plant responses to climate warming.

[Response] Thank you very much for this encouraging summary of our work.

[R3_2] It is good to see that two long-term monitoring datasets were used to support their meta-analysis, which confirmed the findings in the study. However, the manuscript needs major revision on the hypotheses and discussion sections.

[Response] Thank you again for your constructive feedback. We have carefully gone through your comments, and addressed each of them accordingly. For more details, please see our point-to-point response to your comments below.

[R3_3] The major flaw of the study is that the expression of hypotheses are too complicated in the introduction and figure 1. I suggest the author to combine the hypotheses into one: For spring phenology: acquisitive species, characterized by high nitrogen concentrations, advance leaf-out more under warming. For autumnal phenology, acquisitive species show more pronounced advancement of senescence in response to warming, while conservative species exhibit opposite response due to thicker leaves and greater carbohydrate reserves. You do not need to break the hypotheses into hypothesis I and II, scenario A scenario B and so on. You can discuss why the hypothesis is testified or not in the discussion in details. In this case Figure 1 need to be changed, only one hypothesis should be shown. The legends of figure 1 needs to be adjusted accordingly. It is annoying to see too complicated hypotheses and confusing to discern whether figure 1 is the hypotheses or the results.

[Response] Thank you very much for pinpointing the core issue of over-complex hypotheses and the confusing structure of Figure 1. We fully agree that our previous hypotheses may be too complicated.

Following your suggestion, we have: 1) Simplified the hypotheses. We removed Scenario A and Scenario B of the autumnal phenology. The revised hypotheses in the **Introduction** section are as follows (**Lines 91-99**): “We test two competing hypotheses (**Fig. 1**): Hypothesis I predicts that acquisitive species, characterized by high nitrogen concentrations, will advance leaf-out more rapidly under warming. These species are also predicted to delay more their autumnal phenology more, likely benefiting from a

warming-induced extension of the photosynthetic window that allows nitrogen-rich plants to fix carbon more efficiently. In contrast, Hypothesis II proposes that conservative species, with thicker leaves and greater carbohydrate reserves, will exhibit a stronger spring advancement. These species are expected to delay autumnal phenology more, likely due to their superior drought tolerance under warming-induced drought.” 2) Modified the Figure 1 and the corresponding legends. Based on the simplified hypotheses, we accordingly modified the Figure 1 and simplified the legend.

Fig. 1 Distribution of the warming experiments and the research hypotheses in this study. **a**, The distribution of the warming experiments and the long-term ground phenological observations are shown. Gray triangles represent warming experiments, and blue and red squares represent data from the USA National Phenology Network (USA-NPN) and the China Phenological Observation Network (CPON), respectively. **b**, **c**, The hypotheses in this study (more details are in the Introduction). Hypothesis I posits that acquisitive species exhibit stronger phenological shifts. Hypothesis II posits that conservative species exhibit greater phenological shifts. ΔPheno , shifts in foliar phenology; Acquisi., acquisitive species; Conser., conservative species.

[R3_4] The discussion can be shorthand and more focused after the hypothesis is adjusted.

[Response] Thank you for your valuable feedback. Following the revision of our hypotheses, we have streamlined the **Discussion** section to improve clarity and focus. The updated discussion now more explicitly addresses how the findings support the specific hypotheses, with a concise explanation of the underlying mechanisms. Additionally, we have minimized repetition of established concepts, such as the widely recognized effect of climate warming on advancing spring phenology, to keep the emphasis on the core conclusions of our study.

Response letter

We appreciate the reviewers' valuable comments and constructive suggestions on the previous version of the manuscript. We have thoroughly addressed all the suggestions made by the reviewers. Below, we list point by point response in the **regular** text next to the all comments/suggestions (in *italics*).

Reviewers' comments to authors

*The line number in this response letter refers to that in the **clean version** of the revised manuscript.

Reviewer #1 (Remarks to the Author):

[R1_1] I feel the authors have sufficiently addressed the concerns I raised in my previous review. I appreciate the pains they took to revise text and figures based on my feedback. I have no further concerns; I feel this paper merits publication.

[Response] We thank you very much for your overall positive evaluation and encouraging comments.

Reviewer #2 (Remarks to the Author):

[R2_1] Thank you for the very comprehensive and successful revision of the paper. My comments have been corrected to my satisfaction. Below are just a few minor suggestions for improvement.

[Response] We thank the reviewer for the positive assessment of our study. We have carefully gone through your suggestions, and addressed each of them accordingly. For more details, please see our point-to-point response to your comments below.

[R2_2] L140: I would delete “also” here.

[Response] We have revised them as suggested (**Line 139**).

[R2_3] L197-199: Is this true for all plants or only for trees? Please be more precise here.

[Response] Thanks for mentioning this issue and we revised this part as (**Lines 198-200**): “Third, plants like temperate deciduous woody species need to fulfill both chilling and heat requirements to break endodormancy and ecodormancy, thereby initiating spring leaf-out^{4,12}.”

[R2_4] Figure 1: I would not change the order of “Acquisitive species” and “Conservative species” in the panels b and c.

[Response] Thanks for this suggestion and we have revised it as suggested.

Fig. 1 Distribution of the warming experiments and the research hypotheses in this study. **a**, The distribution of the warming experiments and the long-term ground phenological observations are shown. Gray triangles represent warming experiments, and blue and red squares represent data from the USA National Phenology Network (USA-NPN) and the China Phenological Observation Network (CPON), respectively. **b**, **c**, The hypotheses in this study (more details are in the Introduction). Hypothesis I posits that acquisitive species exhibit stronger phenological shifts. Hypothesis II posits that conservative species exhibit greater phenological shifts. Δ Pheno, shifts in foliar phenology; Acquisi., acquisitive species; Conser., conservative species. Source data are provided as a Source Data file.

[R2_5] Figure 4: Thanks for revising this figure. However, I still see potential for improvements. You are actually interested in the comparison of models with and without considering traits, right? If so, I would create a panel a (spring phenology) and b (autumnal phenology). Left column of panels could be climate-only projections and the right the projections with the traits. Using this layout it will be easy to compare the two different projections.

[Response] We thank the reviewer for this constructive suggestion. We have revised Figure 4 as suggested.

Fig. 4 Projections of future foliar phenological changes in terrestrial ecosystems based on climate-only and traits integration projections under two climatic scenarios. **a**, Projected distribution of potential shifts in spring

and autumnal phenology by 2100 under the SSP1-2.6 and SSP5-8.5 scenarios using the climate-only projections. **b**, Projected distribution of potential shifts in spring and autumnal phenology by 2100 under the SSP1-2.6 and SSP5-8.5 scenarios using the traits integration projections. Gray indicates areas with missing data. Each panel incorporates the variation of foliar phenological sensitivity across latitudes. SSP, Shared Socioeconomic Paths. Δ phenology, shifts in foliar phenology. Source data are provided as a Source Data file.

Reviewer #3 (Remarks to the Author):

[R3_1] The authors have well responded to reviewers questions, I think the manuscript is ready for publication.

[Response] Thank you very much for this encouraging summary of our work.